# Eosinophil Responses at the Airway Epithelial Barrier during the Early Phase of Influenza a Virus Infection in C57BL/6 Mice

**DOI:** 10.3390/cells10030509

**Published:** 2021-02-27

**Authors:** Meenakshi Tiwary, Robert J. Rooney, Swantje Liedmann, Kim S. LeMessurier, Amali E. Samarasinghe

**Affiliations:** 1Division of Pulmonology, Allergy-Immunology and Sleep, Department of Pediatrics, College of Medicine, University of Tennessee Health Science Center, Memphis, TN 38163, USA; mtiwary@uthsc.edu (M.T.); rrooney1@uthsc.edu (R.J.R.); klemessu@uthsc.edu (K.S.L.); 2Children’s Foundation Research Institute, Memphis, TN 38105, USA; 3Department of Immunology, St. Jude Children’s Research Hospital, Memphis, TN 38105, USA; Swantje.Liedmann@stjude.org; 4Department of Microbiology, Immunology and Biochemistry, College of Medicine, University of Tennessee Health Science Center, Memphis, TN 38163, USA

**Keywords:** adhesion molecules, cell damage, activation, migration

## Abstract

Eosinophils, previously considered terminally differentiated effector cells, have multifaceted functions in tissues. We previously found that allergic mice with eosinophil-rich inflammation were protected from severe influenza and discovered specialized antiviral effector functions for eosinophils including promoting cellular immunity during influenza. In this study, we hypothesized that eosinophil responses during the early phase of influenza contribute to host protection. Using in vitro and in vivo models, we found that eosinophils were rapidly and dynamically regulated upon influenza A virus (IAV) exposure to gain migratory capabilities to traffic to lymphoid organs after pulmonary infection. Eosinophils were capable of neutralizing virus upon contact and combinations of eosinophil granule proteins reduced virus infectivity through hemagglutinin inactivation. Bi-directional crosstalk between IAV-exposed epithelial cells and eosinophils occurred after IAV infection and cross-regulation promoted barrier responses to improve antiviral defenses in airway epithelial cells. Direct interactions between eosinophils and airway epithelial cells after IAV infection prevented virus-induced cytopathology in airway epithelial cells in vitro, and eosinophil recipient IAV-infected mice also maintained normal airway epithelial cell morphology. Our data suggest that eosinophils are important in the early phase of IAV infection providing immediate protection to the epithelial barrier until adaptive immune responses are deployed during influenza.

## 1. Introduction

As granulocytes armed with a number of cationic proteins that can be toxic to pathogens [1], eosinophils are important in immune defense against large extracellular organisms such as parasites [2]. In the local immunity and/or regulation or repair (LIAR) hypothesis, Lee and colleagues proposed that eosinophil recruitment into tissues may have broader implications in health and disease [3]. Now, eosinophil functions are recognized to be more expansive than mere ‘immune grenades’, including regulation of tissue development, homeostasis, repair [4], and sophisticated host defense against a variety of pathogens [5]. Eosinophils are best-known as culprits in allergic diseases including asthma, primarily due to their recruitment into the lungs and because their elimination with corticosteroids or biologics can lead to symptom alleviation [6]. However, as producers of a plethora of cytokines, eosinophils can regulate T- [7,8], B- [9,10], and mast- cell [11] functions through crosstalk, extending the possibility that the alleviation of asthma symptoms upon their depletion may be due to reduced leukocyte responses. Although respiratory viruses can cause asthma exacerbations, during the 2009 Swine Flu pandemic, asthmatics had diminished morbidity (shortened length of hospital stay, less intensive care unit admissions, less need for mechanical ventilation, less likely to develop secondary bacterial infections) and were less likely to die from influenza compared to non-asthmatics [12], indicating that exacerbations may be host-protective against respiratory viruses. Interestingly, early research surrounding the current coronavirus disease (COVID-19) pandemic also indicates that asthma may not be a co-morbidity associated with severe COVID-19 or mortality [13,14,15]. Together with previously established antiviral functions of eosinophils during respiratory syncytial virus (RSV), rhinovirus (RV), and parainfluenza virus (PIV) infections (reviewed in [5]) it is likely that eosinophils in an established allergic immune milieu may be broadly inimical to respiratory viruses.

Influenza is an infectious disease of the respiratory system that has claimed millions of lives over the past century. Although antiviral therapies and vaccines are now available, influenza viruses are a moving target for the immune system, as well as for the pharmaceutical industry, as they evolve through antigenic drifts, shifts, and re-assortment [16], and continue to be a major public health threat. Intriguingly, a pooled analysis of clinical reports showed that asthmatics fared better during the Swine Flu pandemic of 2009 [12,17,18,19,20,21]. We effectively modeled this phenomenon in mice to show that allergic animals have altered pathophysiologic and immune responses at both the early and late phases of IAV infection [22]. Allergic animals clear the virus sooner than their non-allergic counterparts and have more robust (and specific) CD8^+^ T cell responses [22]. This led us to investigate a possible link between the eosinophil-dominant early immune response and enhanced cellular immunity, and uncover a novel function for eosinophils as mediators of enhanced antiviral CD8^+^ T cell responses [23], which included epigenetic modifications [24]. Based on these findings, we hypothesized that eosinophils undergo temporal phenotypic and functional changes in response to IAV that license their retrograde migration out of the lungs, where the infection occurs, to the draining lymphoid organs to maximize cellular immune defenses, including the activation of CD8^+^ T cells [23]. Here, we investigated eosinophil responses to early stages of IAV infection using our mouse model in the context of migration and trafficking, barrier protection, and virus inhibition (Figure 1A,B) utilizing both in vitro and in vivo platforms. We found that eosinophil activation and adhesion molecules were dynamically regulated when exposed to IAV, and that eosinophils migrated out of the lungs efficiently to lymphoid organs and also participated in regulating epithelial barrier responses to mitigate influenza pathogenesis.

## 2. Materials and Methods

### 2.1. Ethics Statement

All animal work described in this manuscript were approved by the Institutional Animal Care and Use Committee (IACUC, protocol numbers 18.008.0 and 529) at the University of Tennessee Health Science Center and St. Jude Children’s Research Hospital in Memphis, TN, USA.

### 2.2. Animals

Six-week-old C57BL/6J, BALB/cJ, and ΔdblGATA sex-matched mice were purchased from Jackson Laboratories (Bar Harbor, ME, USA) and maintained in micro-isolator cages with alpha-dri bedding and unrestricted access to food and water. Bones were collected from B6;129-Myctm1Slek/J (GFP labelled mice from Jackson Labs) for eosinophil differentiation for adoptive transfer experiments as detailed below. The cages were supplied with purified air and housed in a room with controlled temperature and humidity on a 12 h light–dark cycle.

### 2.3. Viruses and Epithelial Cells

Two strains of H1N1 influenza A virus were used in these experiments. The pandemic (p)H1N1 strain, A/CA/04/2009 (original stock generously provided by Richard Webby, St. Jude) was propagated in Madin–Darby canine kidney.2 (MDCK.2) cells (ATCC, Manassas, VA, USA) and the laboratory strain, A/PR/08/1934 was propagated in embryonated chicken eggs. Hemagglutinin and neuraminidase of both strains of viruses were sequence verified to be devoid of mutations prior to freezing down of large stocks. As our isolate of pH1N1 does not generate plaques on MDCK/A549 cells, we use the tissue culture infectious dose 50% (TCID_50_) method for viral titer determination. Owing to expression of both α-2,6 and α-2,3 linked sialic acid residues, A549 human type I alveolar cell line derived from a carcinoma patient (ATCC) is suitable and has been used to study the pathogenesis of influenza viruses for years [25,26].

### 2.4. Mouse Model of Asthma and Influenza Comorbidity and Tissue Harvest

Fungal antigens are common allergens that asthmatics are sensitized to. Owing to the ubiquity and clinical relevance of *Aspergillus fumigatus* to clinical asthma [27,28], we chose a fungal asthma model for our studies. The induction of allergic asthma was performed by using *A. fumigatus* conidia according to our standard lab protocol as described previously [29], and then infected with 1000 TCID_50_ of pH1N1 virus one week following the second fungal challenge (Figure 1A) as previously described [22]. Recognizing that mice do not develop clinical asthma, the term ‘asthma’ is used loosely here to describe mice that depict the characteristics of allergic disease. The ‘asthma’ control (Ctr) mice were not infected with virus whereas the ‘Flu’ control mice were not subjected to the allergen model but were infected with pH1N1 virus. Naïve mice were neither modelled for asthma nor received the virus infection. We have extensively characterized mucosal and systemic immune profiles in our asthma and Flu co-morbidity models and have established that co-morbid mice have a mixed cytokine profile [22,24,30,31,32].

In order to remove as much contaminating leukocytes as possible prior to staining, lungs and spleens were harvested and digested using gentleMACS dissociator (Miltenyi Biotec, Germany) and single cell suspensions from lungs and spleens were overlaid on 1.084 ρ Ficoll-paque solution (GE Healthcare, Spain) and centrifuged at 960× *g* for 30 min at 20 °C as previously described [22]. Cells were obtained from buffy coat and pellets were stained for flow cytometry with the following antibodies: CD11b (EF450; Invitrogen, Carlsbad, CA, USA), CCR3 (FITC; Biolegend, San Diego, CA, USA), CD62L (BV605; Biolegend), CD69 (APC-Cy7; Biolegend) Influenza A PB-1 (Invitrogen) conjugated with PE (Abcam), Siglec-F (PE-CF594; BD Biosciences, San Jose, CA, USA), ICAM-1 (APC; Biolegend), VLA-4 (PE-Cy7; Biolegend). Unstained cells, single color controls, and isotype controls were used for the cytometer set up for each experiment and to determine negative populations. Data were acquired using a BD LSR Fortessa and analyses using FlowJo v10.5.2 (Treestar, Ashland, OR, USA) software.

### 2.5. Generation of Mouse Bone Marrow-Derived Eosinophils (BMdEos) and Exposure to IAV

Bone marrow harvested from tibias and femurs of mice were used to derive eosinophils as detailed elsewhere [33]. All recombinant proteins were purchased from PeproTech (Rocky Hill, NJ, USA). BMdEos maintained on media supplemented with rmIL-5 were exposed to PR8 at 1 multiplicity of infection (MOI) while cells in the control group were mock-infected as previously described [24]. Mock and infected cells were stained with the same antibodies as listed above to identify activation and adhesion markers on eosinophils.

For imaging of eosinophils in vivo, bone marrow-derived eosinophils were generated as described above from bones of B6-GFP-c-Myc mice (Jackson Labs). Eosinophils were adoptively transferred intratracheally into pH1N1-infected mice and mediastinal lymph nodes (MLNs) were harvested 24 h later and fixed in PBS containing 2% PFA, 0.1% T-100, and 1% DMSO overnight at 4 °C. Tissues were cryosectioned onto charged glass slides, blocked in PBS containing 1% BSA and incubated with anti-CD3ε (sc-1127, Santa Cruz Biotechnology, Dallas, TX, USA) at 1:200 dilution overnight at 4 °C. Slides were washed in PBS and incubated for 1 h at RT with 1:1000 diluted donkey anti-goat-AF594 (705-585147, Jackson ImmunoResearch, West Grove, PA, USA). Slides were mounted with Prolong Glass Antifade Mountant with NucBlue Stain (P36983, Invitrogen). For imaging, a Marianas confocal (Intelligent Imaging Innovations) comprised of a CSU-C spinning disk, Prime95B sCMOS camera, and 405, 488, 561, and 640 nm laser lines was used. Images were analyzed using Slidebook 6 imaging software (Intelligent Imaging Innovations).

### 2.6. Determination of Eosinophil Impact on IAV Infectivity

In order to elucidate the impact of eosinophil granule proteins on virus infectivity, 100,000 TCID_50_ units of pH1N1 was incubated for 1 h in the presence of 1 µg/mL of each granule protein (MyBioSource, San Diego, CA, USA) individually or in combination. Supernatants were used to determine the HA titer and virus titer in comparison to media-exposed virus.

To determine whether eosinophil-mediated reduction in virus infectivity affected viral virulence in vivo, BMdEos were exposed to 20,000 TCID_50_/mL of PR8 at MOI of 0.01 and incubated for 1 h at 37 °C with 5% CO_2_. Cells were pelleted at 3500× *g* for 10 min and mice were infected with 50 µL of the supernatant by intranasal route. Virus that was pre-incubated in growth media without cells were used to infect control mice. Mice were euthanized 1- and 3-days post infection and lungs were harvested and stored at −80 °C until use. Lungs were homogenized in PBS containing protease inhibitor cocktail (Roche, Basel, Switzerland). Viral load was determined as previously described [23].

### 2.7. Adoptive Transfer of Eosinophils and Determination of the Airway Epithelial Cell Height

Eosinophils were harvested from the lungs of allergic mice and transferred into pH1N1-infected mice as previously described [23] to determine whether eosinophils impact the integrity of airway epithelial cells. Eosinophil deficient mice (ΔdblGATA) and wild-type control (BALB/cJ) mice were subjected to the acute asthma and influenza co-morbidity model as previously described. Lungs were harvested, formalin-fixed, and hematoxylin and eosin (H&E) and periodic acid Schiff’s (PAS) stains were performed on 4 µm-thick sections of lungs. Five large airways were photographed at 200× and 400× using a Nikon Eclipse upright light microscope (Melville, NY, USA) on each lung specimen and the height of 10 bronchial epithelial cells on each airway were measured by NIS Elements Software (Nikon, Tokyo, Japan). The mean height of epithelial cells was calculated with standard deviation for each group.

### 2.8. Determination of the Impact of Virus Infection on Cell Activation

We used A549 cells, a transformed human cell line, in these studies as a proxy for lung alveolar epithelial cells. We maintain large stocks of A549 cells at low passage and verify that they are free from mycoplasma contamination routinely. A549 cells were seeded in 24-well plates and incubated for 24 h at 37 °C with 5% CO_2_. Confluent monolayers of A549 cells were infected with 0.5 MOI of pH1N1 and cells were incubated at 37 °C with 5% CO_2_ in the presence of 1 µg/mL of TPCK-trypsin. Mock-infected cells received infection media instead of virus. Mature BMdEos suspended in growth media supplemented with 10 ng/mL of IL-5 and 5 ng/mL of rmGM-CSF were added at 1:1 ratio to A549 cells directly or indirectly by using Transwell^®^ permeable supports (0.4 µM permeable membrane) (Castor, Kennebunk, ME, USA). Plates were incubated for 3 days at 37 °C and 5% CO_2_ and cells were individually stained for flow cytometry. A549 cells were stained with anti-human HLA-A2 (PE-Cy7; Biolegend), CD40 (BV421; Biolegend), CD69 (A700; Biolegend), and influenza A PB1 (PE). Eosinophils were stained with anti-mouse MHC-I (PE-Cy7; Biolegend), CD80 (BV605; Biolegend), Siglec-F (PE-CF594, BD Biosciences), CD69 (APC-Cy7; Biolegend), and influenza A PB-1 (PE). Staining and acquisition were performed individually on each cell type with a BD LSR Fortessa and analyzed using FlowJo v10.5.2 (Treestar) software.

### 2.9. Microarray Gene Expression Profiling of Epithelial Cells

Confluent monolayers of A549 cells were infected with 0.5 MOI of pH1N1 or mock-infected, and co-cultured directly (d/D) or indirectly (id/ID) with BMdEos, as described above, or mock controls for three days at 37 °C with 5% CO_2_. BMdEos cells were separated out and A549 cells were used to isolate RNA with the RNeasy Mini RNA extraction kit (Qiagen, Hilden, Germany) per the manufacturer’s recommended procedure. RNA samples were quantified and analyzed for RNA integrity using an Agilent 2100 bioanalyzer (Agilent Technologies, Santa Clara, CA, USA). Microarray analysis services were provided by the ThermoFisher Microarray Research Services Laboratory (Santa Clara, CA, USA) and used Clariom S Assays HT for human samples, according to the manufacturer’s recommended procedures (Affymetrix, Santa Clara, CA, USA). Microarray data has been deposited into the Gene Expression Omnibus (GEO) database under GEO submission GSE163224.

Microarray expression values were generated using the SST-RMA method. Gene-level expression values were filtered prior to statistical analysis for category = main; chr = mapped; and a minimum value ≥ 4.00 in at least one treatment group, and then were subjected to 2-way ANOVA (2 independent variables: IAV infection, BMdEos exposure) with Westfall-Young (W-Y) false discovery rate correction, and independent *t*-tests with Benjamini–Hockberg (B-H) false discovery rate correction for group vs. group comparisons. Criteria for differentially expressed genes (DEGs) in the IAV-infected (no BMdEos) vs. mock-infected (no BMdEos) comparison (V vs. M) were: (1) any W-Y adjusted ANOVA *p*-value ≤ 0.05; and (2) an absolute log_2_ fold change value ≥ 1.00 and a B-H adjusted *t*-test *p*-value ≤ 0.05 for the V vs. M group comparison. DEGs in the V vs. M group comparison were then filtered to identify DEGs influenced by direct BMdEos exposure (VD vs. V comparison) or indirect BMdEos exposure (VID vs. V comparison) according to the following criteria: (1) 2-way W-Y adjusted ANOVA *p*-value ≤ 0.05 for BMdEos exposure or for Interaction between IAV infection and BMdEos exposure; and (2) an absolute log_2_ fold change value ≥ 0.585 (1.5 fold change) and a B-H adjusted *t*-test *p*-value ≤ 0.05 for the respective group comparison (VD vs. V or VID vs. V).

Unsupervised hierarchical clustering and heat map generation were performed in TM4 MeV [34] using log_2_ transformed signal values row centered to the M group mean. Probe set clustering was by complete linkage based on Euclidean distance as the similarity metric. Pathway enrichment analyses using KEGG, Panther, Reactome, and Wikipathway databases and transcription factor target enrichment using the mSigDB database were performed in Webgestalt [35], and protein interaction analyses were performed in STRINGdb [36], based on the indicated lists of DEGs encoding identified proteins.

### 2.10. Statistical Analyses

All the animal work and in vitro experiments were performed with 5–6 mice and 5–6 replicate wells, respectively, in each group for rigor. All experiments were independently repeated at least twice for reproducibility. Data are represented as mean and standard deviation (SD). The determination of statistical significance was done using GraphPad Prism software v6.05 (La Jolla, CA, USA) and tests used are noted in the Figure Legends. Significance values of *p* < 0.05 are marked by asterisks (*) or significantly different groups are denoted with different letters above bars.

## 3. Results

### 3.1. Eosinophils Are Activated by Influenza a Virus In Vivo

While not directly investigated, eosinophils have been found in mouse lungs [37] and human blood [38] during influenza. We previously reported that ~25% of the cells found in the airways in AA+Flu mice were eosinophils compared to ~5% in Flu Ctr animals [22], and that these cells were active within airways even in the absence of an allergic stimulus [24]. However, what impact IAV infection has on the eosinophils that are poised in the lungs at the time of infection was unknown and served as the impetus for this work. We investigated eosinophil activation and adhesion molecule expression in the lungs and spleens as shown in Figure 1A to begin testing our hypothesis that eosinophils directly inhibit virus pathogenesis while protecting the epithelial barrier and activating adaptive immune responses (Figure 1B).

Using the gating strategy in Figure 1, we measured the number of eosinophils in the lungs and spleens and determined the frequency of eosinophils that expressed viral antigen PB-1. We then focused on these infected cells to identify changes to major surface markers important for inflammation. As expected, low eosinophil numbers were found in the lungs (2.41 × 10^4^ ± 8.4 × 10^3^ viable cells) and spleens (3.97 × 10^4^ ± 1.35 ×10^4^ viable cells) of naïve mice. Eosinophil influx into lungs was increased as a result of IAV infection, although Asthma Ctr mice maintained higher numbers of eosinophils in the spleen compared to other groups throughout the time course (shaded areas Figure 1C,D). Approximately 2–60% of eosinophils in the spleens and lungs were infected with IAV as indicated by surface expression of viral PB-1 antigen on eosinophils in both groups (Figure 1C,D). Since the lungs are the site of infection, eosinophils that expressed PB-1 within the spleens most likely migrated from the lungs, in which case, they would need to alter their activation and adhesion molecules which we analyzed next.

PB-1^+^ eosinophils in the lungs had altered expression of CD62L (L-selectin) and CD69. Eosinophils in the lungs of both treatment groups expressed higher levels of CD62L until a rapid reduction at day 7. The majority of infected eosinophils in the lungs expressed CD69 peaking at days 3 and 5 in both groups (Figure 1C). In the spleen, fewer eosinophils expressed CD62L but nearly all eosinophils in both groups expressed CD69 (Figure 1D). Some eosinophils in the lungs of both groups expressed ICAM-1 throughout the course of infection. VLA-4 expressing eosinophils were more prominent during the early time points after infection albeit expression was in >25% of the eosinophils in the lungs (Figure 1C). Most eosinophils in the spleens of both groups expressed ICAM-1 at day 1 with a gradual reduction over time, while VLA-4 expressing eosinophils peaked at day 3 in both groups with near complete reduction after that (Figure 1D). While some statistical differences were noted between treatment groups, overall trends were equivalent between groups suggesting that eosinophils within the lungs and spleens were responsive to IAV infection, and that activated eosinophils expressed markers necessary to migrate into the lymphoid organs from the site of infection 3.

### 3.2. Eosinophil Phenotypic Responses to IAV Are Temporally Regulated

Innate cell activation in situ in response to antigens and cytokines affords eosinophils the opportunity to interact with neighboring structural cells and other immune cells. While the expression of cytokine/chemokine receptors is important for cellular crosstalk, the regulated expression of integrin/lectin molecules is also an important determinant to migratory properties of eosinophils. Given our previous finding that eosinophils are activated by IAV [24] and current data that eosinophils alter their surface markers important for cell activation and migration in vivo (Figure 1), we investigated the temporal regulation of these events in vitro by analyzing cells exposed to PR8 virus for pre-determined times (Figure 2A).

Virus-exposure caused time-dependent changes in cell activation (Figure 2B) and adhesion (Figure 2C) markers. CD62L is expressed on eosinophils at baseline and its downregulation is considered a marker of cell activation [39]. Virus-exposure caused an immediate upregulation of CD62L expression, which was equivalent to mock-exposed cells at 1 h and gradually decreased over time in both groups (Figure 2B). CD69 and CD11b are upregulated on lung and blood eosinophils in asthmatics [40,41,42] and in response to allergen stimulation in mice [43], although they have not been investigated in the context of respiratory viruses. Both markers were rapidly upregulated in IAV-exposed eosinophils; however, while CD69 expression remained markedly higher on IAV-exposed eosinophils, CD11b expression was higher in mock-exposed cells (Figure 2B).

Adhesion molecules are multifunctional as they guide cell movement along blood vessel endothelia [44], permit cell trafficking within tissues [45], as well as forming anchors that sanction immune synapses [46]. IAV exposure caused alterations to the surface expression of two major adhesion molecules on the surface of eosinophils (Figure 2C). While ICAM-1 upregulation occurs as an immediate response to IAV and remains elevated once expressed, VLA-4 expression was transiently downregulated after IAV exposure and upregulated at 6 h (Figure 2C).

In order to test whether eosinophils in the airways transmigrate the bronchial epithelium and drain into the lymph nodes, we adoptively transferred eosinophils derived from GFP-tagged mice into virus-infected mice intratracheally and visualized their presence in the MLNs. We noted their presence in the MLNs and observed that they localized to the T cell zones of the MLNs (Figure 2D) and spleens. Together, these data suggest that eosinophil responses to IAV are immediate and include the ability to migrate in a retrograde manner.

### 3.3. Eosinophils Reduce Virus Infectivity and May Directly Contribute to Bronchial Barrier Protection

Eosinophils undergo piecemeal degranulation in response to IAV [23], and eosinophil granule proteins are recognized inhibitors of respiratory virus infectivity [47,48]. In order to determine what impact eosinophil granule proteins have on IAV, we exposed influenza virions to eosinophil granule proteins either independently or in combination. Individually, major basic protein (MBP) and RNases 2 and 3 did not affect the activity of sialic acid binding protein hemagglutinin (HA) (Figure 3A) nor the virus’ ability to infect epithelial cells (Figure 3B). Both RNases together caused a significant reduction in the HA titer and the virus load in epithelial cells, with MBP inhibiting IAV hemagglutination further when added in combination with both RNases (Figure 3A,B). Although MBP and RNase3 together led to a reduction in the HA titer (Figure 3A), the reduction in virus titer did not reach statistical significance (Figure 3B). Concordant with these granule protein treatment data, incubating log_10_4.3 TCID_50_ IAV at a 1:1 ratio with eosinophils (MOI of 1) for 1 h at 37 °C led to a 100-fold reduction in viral titer (log_10_2.28 ± 0.44 TCID_50_), which was a significant decrease (*p* < 0.01, by Mann–Whitney test). Because these results suggested that eosinophils can inhibit virus infectivity, we determined their in vivo relevance next. Recipients of eosinophil-exposed viruses had reduced virus load compared to mice that were inoculated with the media-exposed pH1N1 24 h after inoculation and the viral load remained low when measured at 72 h (Figure 3C) suggesting effective neutralization by eosinophils.

Additionally, as the bronchial epithelia in allergic mice do not show virus-induced damage [22], we conducted an in vivo adoptive transfer of eosinophils from the lungs of allergic mice into virus-infected animals to determine if eosinophils may help protect the epithelial barrier in vivo during influenza (Figure 3D). Neither eosinophils nor IAV infection alone led to goblet cell metaplasia as noted by the absence of magenta stained goblet cells in the epithelial barrier after PAS staining (Figure 3E). While the bronchial epithelial barrier in virus-infected mice that did not receive eosinophils were damaged with cilia loss and resembled squamous epithelial morphology, the normal columnar shape was maintained in the bronchial epithelial barrier in eosinophil recipient mice (Figure 3E). No changes to the epithelial barrier were noted when eosinophils were transferred into naïve mouse lungs (Figure 3E). As previously published [23], the viral load was equivalent between eosinophil recipient and non-recipient mice at early timepoints after infection (data not shown). We measured the height of the large airway epithelial cell lining as a proxy for morphologic changes indicative of the loss of barrier integrity and found that the reduction in epithelial cell height induced by virus infection was alleviated in virus-infected mice that received eosinophils (Figure 3F).

In order to determine if eosinophil infiltration into the lungs during allergic disease helps to safeguard the epithelial barrier integrity during influenza, we compared the airway epithelial lining of wild-type and eosinophil deficient mice subjected to our asthma and influenza co-morbidity model (Figure 3G,H). No differences were observed in the airway lining of naïve animals and those with induced ‘asthma’ (Figure 3G). However, when the epithelial cell heights were measured, mice devoid of eosinophils were noted to be slightly shorter compared to WT mice (Figure 3H). Virus infection-induced damage to these columnar epithelia were also notable in the ΔdblGATA mice (Figure 3G,H). While the epithelial barrier appeared to maintain the columnar shape in the eosinophil deficient co-morbid mice (Figure 3G), cell height was reduced in comparison to the WT controls (Figure 3H). Goblet cell metaplasia was evident in the asthma groups of both WT and ΔdblGATA mice, and virus infection did not alter the number of goblet cells that interspersed the airways in either group of mice (data not shown).

### 3.4. Eosinophils Reduce Virus-Induced Cytopathology in Airway Epithelial Cells

As a cytopathic virus, IAV causes significant damage to airway epithelial cells during its life cycle. Circumstantial evidence that eosinophil degranulation in the tissue can be injurious to the host [49] have implicated eosinophils as offenders in pathologic allergic diseases like asthma [50] and eosinophilic esophagitis [51]. However, our data suggest that eosinophils may alleviate virus-induced damage to the airway epithelia (Figure 3). In order to test the hypothesis that eosinophils are protective to the epithelial lining during IAV infection, we used a co-culture system with A549 cells and BMdEos where cells were either overlaid in direct contact or separated by a transwell. Epithelial cell morphology was not affected by eosinophils when co-cultured (Figure 4A). Infection with IAV caused the typical cytopathology (cell rounding and loss of contact adhesion) in epithelial cell monolayers, but this cytopathic effect was absent when the epithelial cells were in direct contact with eosinophils. Although limited in comparison to individually cultured virus-infected epithelial cells, we noted that infected epithelial cells co-cultured in indirect contact with eosinophils had some cell rounding and enlargement (Figure 4A). As the virus is cytopathic, we measured the population of dead epithelial cells in the culture conditions to determine if eosinophils affected epithelial cell viability during virus infection and found that non-viable epithelial cells were reduced in the presence of eosinophils (Figure 4B) suggesting that eosinophils may inhibit virus-induced cell death pathways in epithelial cells.

As contact with eosinophils protected epithelial cells from virus-induced cytopathology, we next tested if this was a result of altered susceptibility to virus infection. Eosinophils inhibited the expression of viral proteins on the surface of epithelial cells (Figure 4C) and PB-1^+^ eosinophils decreased when the two cell types were physically separated (Figure 4D) suggesting that direct contact between the cells may promote viral dissemination between cell types thereby reducing the virions available to bind to epithelial cells. Co-localization of these two cell types also reduced the virus load (Figure 4E) suggesting that epithelial cell-eosinophil crosstalk is beneficial to safeguard the airway epithelial cells from virus infection and virus-induced cytopathology during influenza.

### 3.5. Epithelial Cell Transcriptome Is Modified in the Presence of Eosinophils during Virus Infection

Rapid alterations to the A549 epithelial cell transcriptome occur in response to virus infection including in populations of genes associated with cell-cell crosstalk [52]. In order to determine if the presence of eosinophils affected the transcriptomic changes in epithelial cells during virus infection, we analyzed DEGs in A549 cells in the co-culture system described above. IAV infection alone stimulated a robust transcriptional response, where 1969 genes that encode known proteins exhibited significant differential expression (1123 up-regulated, 846 down-regulated) relative to mock infection (Figure 5A). Examination of genes that were up/down regulated in comparison to un-infected epithelial cells without eosinophil contact, showed that most changes occurred when epithelial cells were cultured directly in contact with eosinophils (Figure 5B). Protein interaction network (Figure 5C and Appendix A) and pathway network analysis (Appendix A) based on DEGs that were upregulated with IAV infection showed prominent participation and significant over-representation of genes involved in interferon (IFN) and other cytokine signaling pathways, immune system, and inflammatory responses to viruses; cell cycle regulation, DNA replication, damage, and repair; and cell death. Accordingly, many of these DEGs were identified as targets of transcription factors known to participate in these processes and pathways, including IFN regulatory factors (IRF1/2/7), NF-κB, STAT5 and STAT6, E2F factors, OCT factors, NFY, HNF3, E4F1, and ATF factors (Appendix A). By contrast, protein interaction networks (Figure 5D and Appendix A) and pathway analysis (Appendix A) of the protein-encoding DEGs that were down-regulated with IAV infection highlighted processes and pathways related to amino acid, lipid, and carbon metabolism; ATP production (TCA cycle and respiratory electron transport); protein translation; platelet activation and the coagulation cascade; and xenobiotic metabolism. Genes targeted by HNF1, HNF4, and PPAR factors were statistically over-represented in this set of down-regulated DEGs but, overall, significantly fewer of the down-regulated DEGs were identified as specific transcription factor targets (Appendix A). A list of protein-encoding DEGs in the IAV infection vs. mock infection comparison (V vs. M) is provided in Appendix A.

The extent to which eosinophil exposure affected the A549 cell transcriptome varied with infection and the method of exposure (direct or indirect). The largest effect occurred with IAV-infected cells that were directly exposed to eosinophils, where 625 of the IAV-induced protein-encoding DEGs exhibited significant changes in expression (91 up-regulated, 534 down-regulated) when compared to IAV-infected cells that were not exposed to eosinophils (Figure 5A). Protein interaction networks (Figure 5E and Appendix A) and pathway analysis (Appendix A) of the IAV-induced DEGs down-regulated with direct eosinophil exposure highlighted many of the same processes and pathways that were up-regulated during IAV infection (i.e., IFN and other cytokine signaling pathways, immune system, and inflammatory responses to viruses; cell cycle regulation, DNA replication, damage, and repair; and cell death). Notably, the reduction in expression of 240 of these DEGs was sufficient to counteract their IAV-induced increase by 50% or greater. Pathway analysis of the 91 IAV-induced DEGs that were up-regulated with direct eosinophil exposure did not reveal statistically significant over-representation of genes in any particular pathway (not shown), although protein interaction networks highlighted processes involved in ATP production and mitochondrial function (Figure 5F and Appendix A). Direct eosinophil exposure to un-infected A549 cells produced significantly fewer transcriptomic changes than in IAV-infected cells (Figure 5A), however the pathways affected by those changes were generally the same as those affected with direct eosinophil exposure (not shown). A list of protein-encoding DEGs in the comparison of IAV infection with direct BMdEos exposure vs. IAV infection alone (VD vs. V) is provided in Appendix A.

Indirect eosinophil exposure had much less effect on the transcriptome of IAV-infected cells than direct exposure, resulting in 128 DEGs (74 up-regulated, 56 down-regulated) when compared to IAV-infected cells without eosinophils (Figure 5A). Curiously, the 74 IAV-induced DEGs that were up-regulated with indirect eosinophil exposure had an over-representation of genes involved in cell cycle regulation, DNA replication, and DNA repair; and cytokine signaling, which are pathways that were down-regulated with direct eosinophil exposure (Appendix A). However, the 56 DEGs down regulated with indirect exposure had an over-representation of genes involved in amino acid metabolism and protein translation (Appendix A), which is the same trend observed with direct exposure. A list of protein-encoding DEGs in comparison of IAV infection with indirect BMdEos exposure vs. IAV infection alone (VID vs. V) is provided in Appendix A. Indirect eosinophil exposure had an even smaller effect on un-infected A549 cells, resulting in 77 DEGs (55 up-regulated, 22 down-regulated) that showed no significant pathway over-representation (not shown) and no correlation with the transcriptomic effects observed in the other three conditions.

### 3.6. Airway Epithelial Cell—Eosinophil Crosstalk Promotes Activation in Both Cell Types

As our data suggested that eosinophils and epithelial cells interacted during IAV infection, we then assessed if co-regulatory mechanisms occurred between airway epithelial cells and eosinophils during co-culture (Figure 6). Interestingly, the presence of eosinophils caused epithelial cells to reduce MHC-I (HLA) expression, which remained lowered after IAV infection (Figure 6B). Neither CD40 nor CD69 on epithelial cells were affected by the presence of eosinophils in the absence of IAV infection, but epithelial cells that were in contact with eosinophils increased expression of CD40 and decreased CD69 expression after virus exposure. These changes in IAV-exposed cells were more apparent when the eosinophils were separated by transwell (Figure 6B).

Eosinophils also altered expression of surface markers important for antigen presentation and activation. When placed in direct contact with un-infected epithelial cells, eosinophils increased expression of MHC-I and CD69 and reduced CD80 (Figure 6B). Eosinophil antigen presenting markers were reduced when co-cultured with infected epithelial cells, but when the cell types were separated by transwell, eosinophils did not regulate these markers in response to IAV (Figure 6B). Cumulatively, these data indicate that eosinophil-epithelial crosstalk can occur when placed in close proximity during IAV infection.

## 4. Discussion

Over a century has passed since the identification of eosinophils, yet their role in the immune system and reasons for evolutionary conservation continue to be a topic of debate. By supporting a more comprehensive role for eosinophils with functions in tissue development, homeostasis, defense, and repair, the LIAR hypothesis [3] antiquates the viewpoint that eosinophils are simply the host’s antiparasitic defense strategy. Although eosinophils are not among the primary arsenal of leukocytes that defend against respiratory pathogens, their ability to release antimicrobial molecules [53] and enlist other leukocytes, such as dendritic cells [54,55,56] and T cells [8], provide a rationale to investigate their immunophenotypic and functional responses in various milieus to gain a better understanding of the immunomodulatory functions during pulmonary infections. Here, we examined eosinophil responses to IAV during the early phase of influenza and found that eosinophils exhibit multiple functions as active mediators of antiviral host defense through virus neutralization, trafficking to draining lymphoid organs, and protecting the airway barrier from virus-induced cytopathology.

Eosinophil granule proteins have been shown to hinder infectivity of RSV, where RNases 2 and 3 were potent at virus neutralization [47,57]. Here, we show that while individually ineffective, eosinophil RNases 2 and 3 when combined are capable of inhibiting IAV infectivity. As individual granule proteins did not affect IAV infectivity, it is possible that whole granule content release through other degranulation pathways may be necessary for this outcome. Intriguingly, since eosinophil neutralization of PIV is dependent on their ability to generate nitric oxide rather than through granule proteins [58,59], it is possible that a similar pathway may be active during IAV infections. It is also worth noting that MBP was individually ineffective at IAV neutralization. Although eosinophils did not injure A549 cells in our studies, high concentrations of MBP damaged RSV-infected A549 cells [60,61]. Together, these data suggest that MBP may have alternative functions during IAV infection in vivo and warrants further investigation.

When considered along with our previous findings that eosinophils undergo piecemeal degranulation in response to IAV [23] and that eosinophil peroxidase levels are elevated in the bronchoalveolar lavage fluid and lung homogenates in mice with asthma and influenza co-morbidity compared to asthma alone (unpublished data), the current results suggest that locally released granule proteins can act in concert to hinder IAV infectivity. The immediacy of this eosinophilic function on IAV in vivo, however, must still be determined since our previous study showed no detectable difference in viral load between allergic and non-allergic animals nor between eosinophil recipient and non-recipient virus-infected mice at early times after IAV infection [22]. There may be several explanations behind this discordance. Although eosinophils are clearly capable of neutralizing virus in close proximity, physical barriers such as increased mucus in the airways of allergic hosts [22] may prevent the juxtaposition of eosinophils and infected epithelial cells, limiting the opportunity for eosinophils to directly inhibit IAV early on. It is also possible that IAV adsorption to epithelial cells occurs before eosinophil localization to the epithelial barrier. The presence of PB-1^+^ eosinophils in the spleen together with eosinophil expression of adhesion molecules after IAV exposure strongly suggest that at least some proportion of eosinophils migrate out of the lungs after virus infection which would also limit the in situ neutralization of IAV. Finally, as in vitro assays are devoid of other leukocytes (and secretory products) and lack the intricate lung architecture comprised of other structural cells and the extracellular matrix, the possibility that eosinophil neutralization of virus seen in this assay is an in vitro artefact cannot be dismissed out of hand. In light of these possibilities, in vivo studies utilizing fluorescent-labeled IAV to determine the in situ localization of the virus in relation to eosinophils in real time in addition to the determination of mediators released during piecemeal degranulation of IAV-exposed eosinophils and the kinetics thereof are necessary to gain insight into establishing the immediate effector functions of eosinophils within the airways and lungs of mice during the early phases of infection.

The pathogenesis of influenza is largely due to a combination of virus replication-induced damage and cytotoxic immune responses, and our data suggest eosinophils can act in ways to mitigate this damage. Airway epithelial cells are the primary target of IAV and virus replication has been shown to have several effects on the epithelial barrier [62], including the induction of various cell death pathways [63] that can kill as many as 50% of infected epithelial cells within three days of infection [64]. Here, we found that in addition to reduced susceptibility to infection, as well as a reduction in viral titer, epithelial cells exhibited significantly lower levels of IAV-induced cell death in the presence of eosinophils. Virus-infection induced epithelial cell death can alter the barrier integrity thereby affecting a number of downstream pathologies associated with IAV infection in the host [65], including bacterial superinfections [62,66]. Epithelial cell death was reduced in the presence of eosinophils possibly through the altered regulation of genes associated with the extrinsic pathway of apoptosis, such as *FAS*, *CASP7*, *PARP*, *LAMA* [67], found to be upregulated in the IAV-infected epithelial cells (V group) but downmodulated in the IAV-infected epithelial cells cultured directly with eosinophils (VD group). X-linked inhibitor of apoptosis associated factor-1 (*XAF1*), an IFN inducible tumor suppressor gene considered to promote apoptosis in lung epithelial and peripheral blood mononuclear cells [68,69], was the most upregulated gene in epithelial cells during IAV infection. The direct co-culture of infected epithelial cells with eosinophils led to a significant downregulation of *XAF1*. Therefore, our data suggest that multiple pathways of cell death in epithelial cells may be inhibited by means of epithelial-eosinophil crosstalk. Previous data showed that allergic mice with eosinophils had significantly less damage to the bronchial epithelial barrier when infected with IAV [22], and the reduction of virus-induced cytopathology in the presence of eosinophils observed here, both in the co-culture setting and in eosinophil recipient mouse lungs, similarly supports a barrier protective role for eosinophils.

Although human eosinophils and their granule products have long been considered cytotoxic [70] and RSV-infected epithelial cells are damaged when exposed to eosinophil granule proteins [60], we did not observe any morphological changes in epithelial cells nor elevated cell death in any co-cultures of epithelial cells and eosinophils in this study. Furthermore, the addition of eosinophil-conditioned media to cultures of IAV-infected epithelial cells also did not have a negative impact (data not shown). Importantly, as demonstrated by Takeda et al. [71], histologic images of lungs from animal models of allergic asthma do not show epithelial damage irrespective of type of allergen, supporting a non-cytotoxic role for eosinophils in wound repair following allergenic challenge. It is not clear if differences in granule protein content and/or mechanisms of degranulation between mouse and human eosinophils, based on the stimuli, are important for the divergent clinical effects of eosinophils observed in these various studies. However, it is clear that eosinophil granules contain a plethora of cytokines, including transforming growth factors, which are important in wound healing processes [72,73]. Thus, as eosinophils undergo piecemeal degranulation in response to IAV [23], cytokines that mitigate the cytopathic effect of IAV may be released to protect the epithelial barrier. Along the same lines, alveolar epithelial cells reduce MHC-I trafficking and cell surface expression during late IAV infection to avoid detection by virus-specific CD8^+^ T cells and thereby survive infection [74]. As IAV-infected A549 had reduced MHC-I (HLA) expression in the presence of eosinophils a correlative observation in vivo may suggest a similar survival response in epithelial cells during influenza. Collectively, our data suggest that eosinophils may defend the airway barrier through epithelial crosstalk, in part by inducing changes in epithelial cells to moderate the extent of cytotoxic damage that would otherwise occur with rampant T cell activity while also reducing virus susceptibility, replication, and apoptosis triggered by viral genes such as PB1-F2 [75].

Changes in the transcriptome of IAV-exposed epithelial cells co-cultured with eosinophils are also consistent with such moderating effects. As previously demonstrated [22,52,76], IAV-exposed epithelial cells showed a robust induction of genes involved in viral RNA sensing and IFN response, cytokine signaling, immune and inflammatory responses, and cell death pathways, whereas the expression of many genes required for normal cellular metabolism, ATP production, and protein translation were reduced. Some mucins are important host defense strategies during influenza pathogenesis [76,77], and we showed that goblet cell metaplasia and associated mucin gene induction occur in the lungs of acute asthma and influenza co-morbid mice in which the airway epithelial lining is intact compared to influenza alone or chronic asthma and influenza models in which airway epithelial cell hyperplasia and necrosis occur [22]. Specifically, mucin genes *MUC13* and *MUC16* were induced by IAV in A549 cells and subsequently downregulated in the presence of eosinophils. Although canonical mucin genes associated with influenza pathogenesis like *MUC1* and *MUC5B* [76,77] were not regulated in A549 cells, and we did note goblet cell metaplasia to occur as a result of eosinophil transfer or differences in the eosinophil-deficient mice (data not shown), it is worth investigating the effect eosinophils have on mucin composition and goblet cell metaplasia in detail in the context of influenza.

Many cell cycle-regulated genes expressed during S/G_2_ phase were also induced in IAV-infected epithelial cells, suggesting arrest at this cell cycle stage is more physiological advantageous for viral pathogenesis; although viral-host protein interactions appear to induce G_0_/G_1_ arrest during the initial rounds of IAV replication [78,79,80,81], S/G_2_ arrest may be characteristic of severe and/or later infection [82]. By contrast, direct co-culture of IAV-exposed epithelial cells with eosinophils restricted or reversed a significant number of these transcriptomic changes, in particular limiting the expression of some genes involved in antiviral, immune and inflammatory response pathways, pro-apoptotic pathways, and S/G_2_ cell cycle arrest, while boosting the expression of five genes required to maintain respiratory electron transport (*ATP5I*, *COA5*, *NDUFA1*, *TOMM7*, *UQCRB*). Indirect co-culture of IAV-exposed epithelial cells with eosinophils had far less influence on such transcriptomic changes, suggesting physical interaction between the two cell types, rather than humoral exposure, may be beneficial. Nonetheless, situations where there is a higher number of eosinophils at the epithelial barrier (such as in the lungs of asthmatics) may provide some protective benefit to the epithelia at the transcriptomic level during an active IAV infection.

Airway inflammation is a major hallmark of respiratory virus infections and uncontrolled inflammation can lead to tissue damage and viral pneumonia. The activated airway epithelium plays a major role as an initiator of airway inflammation during influenza [83,84]. Although our animal models show that allergic animals have heightened inflammation in response to IAV infection, the immune profile significantly differs between allergic and non-allergic mice [22]. When cultured in direct contact with eosinophils, virus-infected epithelial cells downregulated genes associated with type I inflammation like *CCL2*, *CCL22*, *CXCL1*, and *CXCL10* and upregulated *IL-33,* which correspond to our previous in vivo findings [24,32]. These data suggest that eosinophil-epithelial cell interaction during IAV infection can impact the immune milieu thereby affecting the inflammatory profile in the lungs.

Eosinophil activation in response to stimuli is marked by changes to surface markers. Changes to surface markers involved in cellular activation, survival, and migration occur in vivo in response to influenza [24], and here we note that these changes occur rapidly after IAV exposure and correlate with the viral replication cycle in A549 cells [85]. While LFA-1 is the canonical binding partner for ICAM-1, CD11b/CD18 also binds ICAM-1 [86]. In addition to expressing CD11b [87], eosinophils express ICAM-1 and VLA-4 on their cell surface in response to allergen [88] as they did in response to IAV. The intraperitoneal transfer of virus-exposed eosinophils into naïve animals led to the development of IAV-specific T cells, eosinophil presence in the T cell zones of lymphoid organs, and direct interaction with CD8^+^ T cells after virus exposure [23], all suggesting that eosinophils can prime antiviral immune responses. Immune priming requires the formation of immune synapses between the antigen presenter and their cognate T cells. A focal point of the immune synapse is the formation of the supramolecular activation cluster (SMAC) immediately after APC and T-cell interaction. Integrins are also important in the formation of the SMACs during antigen presentation [89], and thus, eosinophil upregulation of ICAM-1 and VCAM-1 as well as their binding partners are likely to play a role in mediating stable interactions with T cells, which were demonstrated to occur in vitro [23].

Eosinophil functions during the early phase of influenza likely involve multiple disparate pathways. These data together with our previous findings [23,24,80] suggest that when pre-poised in the respiratory tract as in the allergic setting, eosinophils may directly neutralize virus (perhaps by piecemeal degranulation), become activated in response to virus infection permitting their retrograde migration out of the lungs into the draining lymphoid organs where they can interact with CD8^+^ T cells in an antigen-specific manner (Figure 7). Additionally, eosinophils at the airway interface may crosstalk with the epithelial barrier by activating them, enhancing their antiviral responses, and protecting epithelial cells from virus-induced damage. Our most recent study demonstrated that eosinophils dynamically respond to IAV by altering surface expression of receptors that allow survival and by slowing down their mitochondrial respiration and upregulating survival markers [24], possibly as countermeasures against host system hijacking and apoptosis induced by IAV [90]. Enhancing their own survival during IAV infection may be a host-protective function during the early phase of influenza. Recognizing that the use of airway epithelial cell line, A549, is a major limitation of our studies, it is important to delineate that similar eosinophil-epithelial crosstalk occurs between bronchial epithelial cells grown in the air-liquid interphase and primary human type I pneumocytes. Additional studies targeted at investigating changes to junctional proteins and cell death pathways in the infected airway epithelia with eosinophils are also necessary to elucidate the mechanisms by which eosinophils counteract virus-induced epithelial damage.

Host–pathogen interactions are affected by the environment including host genetics, microbiome, and local immunity. Another layer of complexity is added when pathogens infect hosts with underlying diseases in which the immune system is already biased. Effective antiviral responses require a strong T_H_1 immune response [91] with significant production of IFNs [92], and asthmatics are more prone toward a T_H_2 response [93] and may have altered IFN production [94]. Therefore, asthmatics have been traditionally considered to be at higher risk for respiratory virus infections [95]. Nonetheless, irrespective of infection susceptibility, asthmatics were less likely to suffer from severe disease—requiring oxygen, mechanical ventilation, developing viral pneumonia, developing secondary bacterial infections, entering the intensive care unit, death—than non-asthmatics during the Swine Flu pandemic of 2009 [12]. We modeled this phenomenon in mice to determine factors that may grant asthmatics protection from severe characteristics associated with influenza despite having an asthma exacerbation and found that eosinophils, which were the most abundant innate leukocyte in the lungs at the time of IAV infection, have a number of antiviral host protective properties that aided the host during early and late phases of influenza. Reports from the ongoing severe acute respiratory syndrome-coronavirus (SARS-CoV)-2 pandemic also have early indicators that patients with allergic asthma are not at increased risk of severe COVID-19 [14,15,96,97,98]. As patients with severe COVID-19 had significantly reduced peripheral blood eosinophils [99,100,101,102], and patients with non-allergic asthma (non-eosinophilic) have increased SARS-CoV-2 receptor expression [103], it is tempting to speculate that eosinophils may play an antiviral role against SARS-CoV-2, similar to their function against RSV, PIV, and IAV. As is true for numerous immune reactions in disease, eosinophil responses to virus infections are likely multifarious and contingent on the host immune bias and the virus strain. Our data support a function for eosinophils during early phases of virus infection that may help protect allergic hosts from severe influenza.

## 5. Conclusions

Eosinophils function as mediators of antiviral host protective mechanisms during the early phase of influenza A virus infection. Their early defense properties may include virus neutralization, retrograde migration into lung draining lymphoid organs following infection, and mediating airway epithelial cell defenses.

## Figures and Tables

**Figure 1 cells-10-00509-f001:**
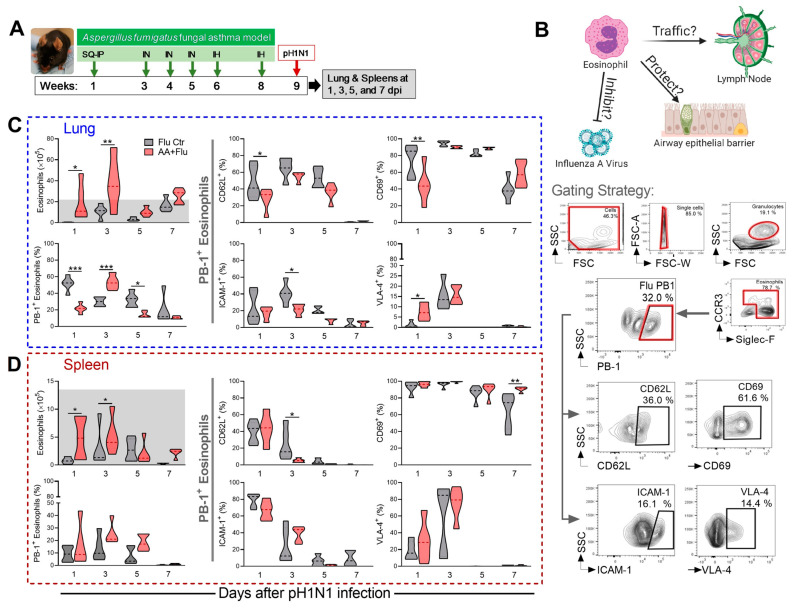
Eosinophils actively respond to influenza A virus infection in vivo. (**A**) Timeline and treatments for mouse model of asthma and influenza comorbidity. (**B**) Schematic representation of proposed hypothesis illustrated in BioRender. Eosinophil quantification and their surface expression of antigens in the (**C**) lungs and (**D**) spleen. Data are represented as the mean and interquartile range with medium values as lines of five mice per group. Experiments were repeated independently to ensure reproducibility. The grey shaded areas show the mean value in the asthma-only group and the dotted lines show the mean value in the naïve group. Between group comparisons were assessed by multiple unpaired *t*-tests (Mann Whitney). * *p* < 0.05, ** *p* < 0.01 and *** *p* < 0.001.

**Figure 2 cells-10-00509-f002:**
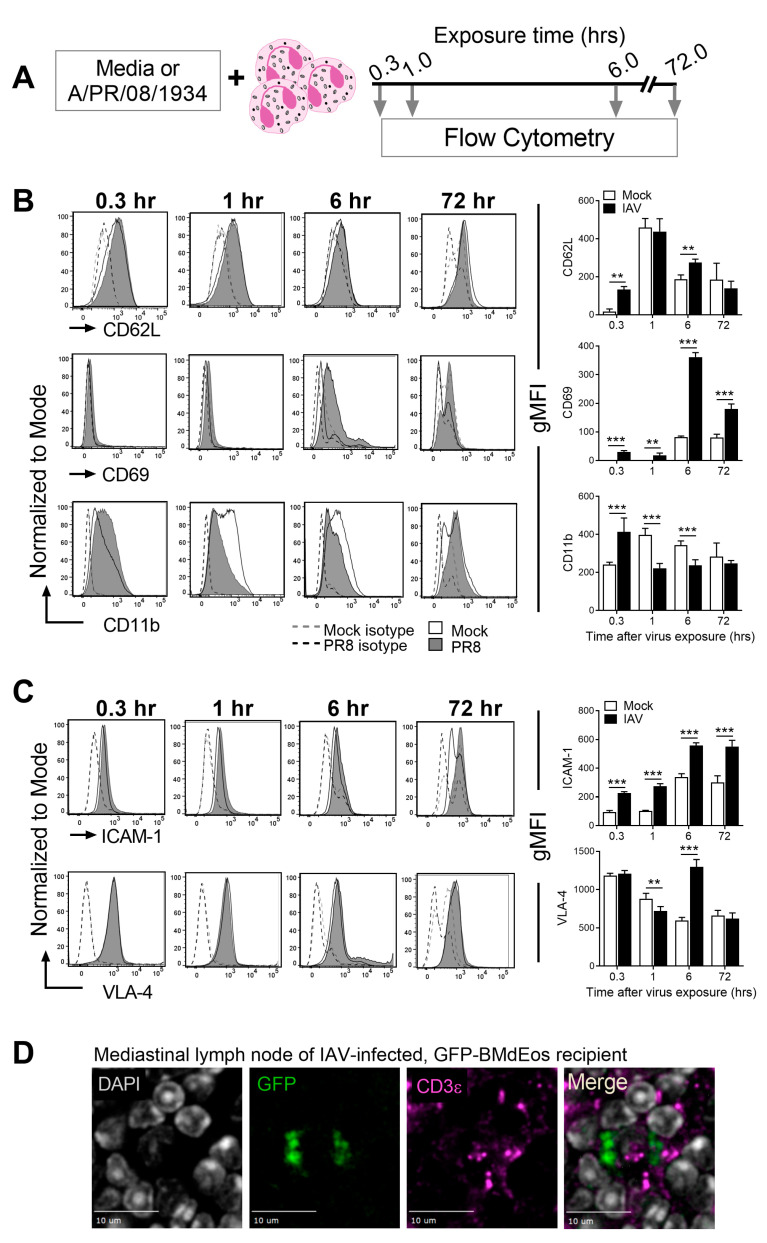
Eosinophils are dynamically regulated after influenza A virus exposure and gain migratory properties. (**A**) Schematic representation of experimental setup. Representative histograms of SSC^hi^SiglecF^+^CCR3^+^ eosinophils from mock (open) and virus exposed (grey) cells and isotype controls (dotted line) and quantification of geometric mean fluorescence intensity (gMFI) for (**B**) activation and (**C**) adhesion markers. (**D**) Green fluorescent protein (GFP)-tagged eosinophils localize to the T cell zones of mediastinal lymph nodes after adoptive transfer into the lungs of virus-infected mice. Experiments were independently repeated three times for reproducibility. Data in graphs are represented as the mean and standard deviation of n = 5 samples analyzed by two-way ANOVA with Tukey’s multiple comparisons test of one independent study. ** *p* < 0.01 and *** *p* < 0.001.

**Figure 3 cells-10-00509-f003:**
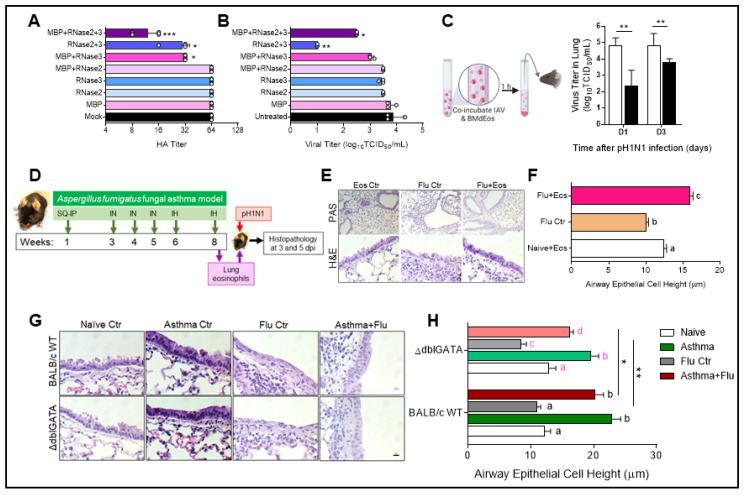
Eosinophils inhibit virus infectivity and protect the airway epithelial barrier from virus-induced cytopathology. Eosinophil-specific granule proteins reduce the (**A**) hemagglutinin (HA) titer and (**B**) virus replication when added in combination. Data are represented as the mean and standard deviation and analyzed with the Kruskal–Wallis multiple comparisons test * *p* < 0.05, ** *p* < 0.01 and *** *p* < 0.001 compared to untreated controls (**C**) Eosinophils and influenza A virus (IAV) incubated at a 1:1 ratio result in reduced virus infectivity in vivo. Data represented as the mean and standard deviation and compared by Mann–Whitney test where ** *p* < 0.01. Illustration with BioRender. (**D**) Schematic representation of adoptive transfer of eosinophils from allergic mice into the lungs of IAV-infected mice. (**E**) Minimal goblet cell metaplasia seen after periodic acid Schiff’s (PAS) stain in response to eosinophil transfer and hematoxylin and eosin (H&E) staining enable the visualization of epithelial barrier integrity shown at day 3. (**F**) Large airway epithelial height measured in each group shown at day 5. (**G**) H&E stains of large airways in each treatment group were used to measure (**H**) epithelial height in wild-type (WT) and eosinophil deficient (ΔdblGATA) mice. Height measurements of bronchial epithelial cells each in five large airways of each lung section. Data shown at day 5. Scale bars = 10 µm in all photomicrographs. Experiments were repeated independently for reproducibility with 5–6 mice/group each time. Data in F and H are represented as the mean and standard error of the mean and analyzed by two-way ANOVA with Sidak’s multiple comparisons test where letter above bars represent * *p* < 0.05 when different and no statistical difference when same.

**Figure 4 cells-10-00509-f004:**
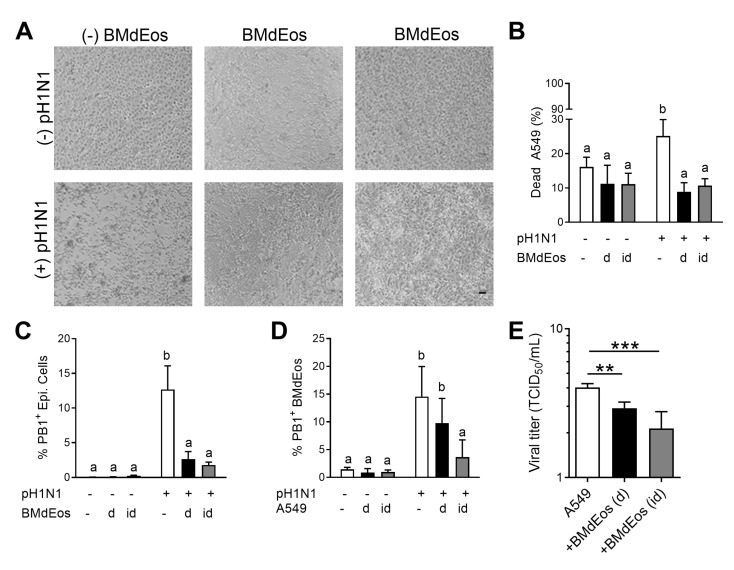
Eosinophils protect the epithelial cell barrier during influenza virus infection. (**A**) Representative images of A549 cell monolayers in each culture condition. (**B**) Percentage of dead A549 cells in the culture conditions. (**C**) Influenza internal protein (PB1) expressing A549 cells and (**D**) bone marrow-derived eosinophils in each culture condition. (**E**) Influenza A virus (pH1N1) titer in the supernatants. Experiments independently repeated three times for rigor and reproducibility. Data are represented as the mean and standard deviation of n = 5–6 samples analyzed by two-way ANOVA with Sidak’s multiple comparisons test. Differences are significant (*p* < 0.05) when letters above bars are dissimilar. ** *p* < 0.01 and *** *p* < 0.001. BMdEos—bone marrow-derived eosinophils; d/D—direct contact; id/ID—indirect contact.

**Figure 5 cells-10-00509-f005:**
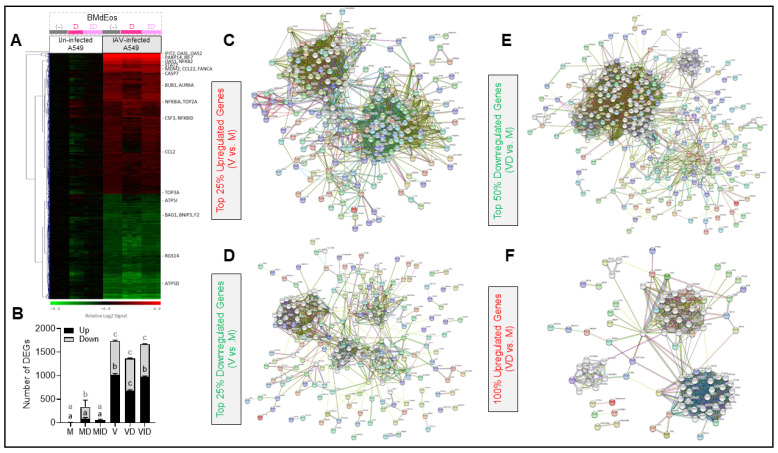
Eosinophils affect the epithelial transcriptome during influenza virus infection. (**A**) Microarray analysis of the A549 cell transcriptome during mock or influenza virus (IAV) infection when in direct (d) or indirect (id) contact with bone marrow derived eosinophils (BMdEos). Heat map contains log_2_ signal values for 1969 individual samples normalized to the row mean log_2_ signal value for samples M1-M4 (Mock-infected, no BMdEos). Dendrogram (left side) shows hierarchical clustering of genes by complete linkage based on euclidean distance. (**B**) The number of differentially expressed genes (DEGs) in each group relative to the average expression in the M group shown as a stacked bar representing the mean and standard deviation. Up- and down-regulated genes were compared across the groups by 2-way ANOVA with Tukey’s multiple comparisons test where letters above bars represent *p* < 0.05 when different. Protein interaction networks generated by STRINGdb are shown for: (**C**) Top 25% of upregulated genes (281 DEGs) between mock-infected and IAV-infected epithelial cells with no BMdEos exposure; network required an edge confidence score ≥0.700 and contains an additional 20 s shell nodes (white). (**D**) Top 25% of downregulated genes (212 DEGs) between mock-infected and IAV-infected epithelial cells with no BMdEos exposure; network required an edge confidence score ≥0.400 and contains an additional 30 s shell nodes (white). (**E**) Top 50% of downregulated genes (267 DEGs) between mock-infected and IAV-infected epithelial cells with direct BMdEos exposure; network required an edge confidence score ≥0.400 and contains an additional 55 s shell nodes (white). (**F**) All upregulated genes (91 DEGs) between mock-infected and IAV-infected epithelial cells with direct BMdEos exposure; network required an edge confidence score ≥0.400 and contains an additional 65 s shell nodes (white). For closer inspection, individual PNG files of the protein interaction networks shown in C-F are provided in Appendix A as Appendix A. Please note that gene expression data and pathway analysis results of these data are provided in Appendix A. M—mock, V—virus, d/D—direct culture, id/ID—indirect culture.

**Figure 6 cells-10-00509-f006:**
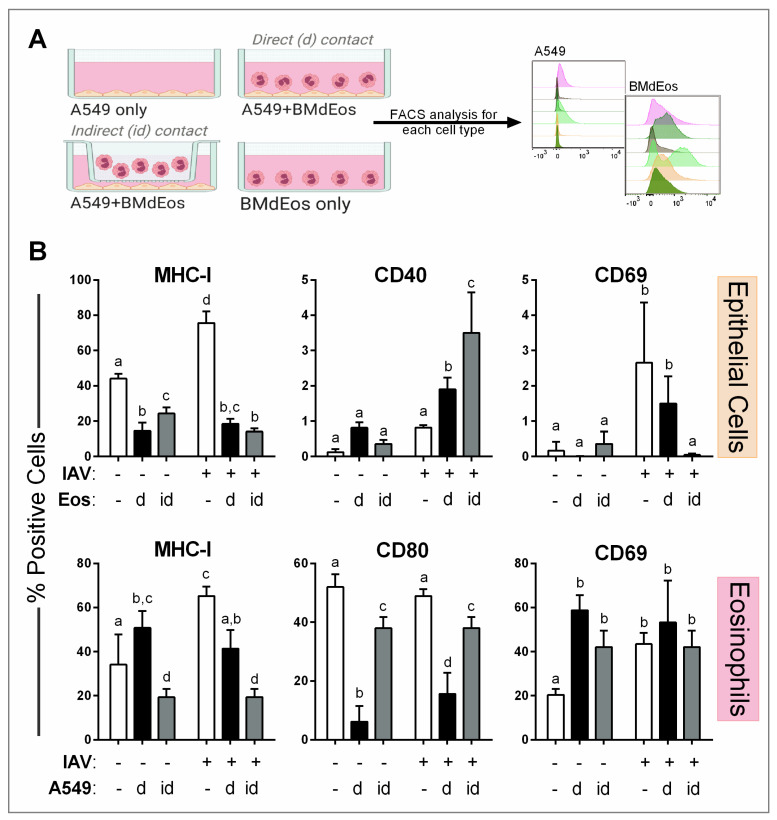
Cross activation of epithelial and eosinophils occurs in response to influenza A virus. (**A**) A549 epithelial cells and bone marrow-derived eosinophils were cultured directly/indirectly after infecting epithelial cells with pH1N1 influenza A virus and each cell type was analyzed by flow cytometry. Illustration with BioRender. (**B**) Surface expression of markers on epithelial cells and eosinophils. Data represented as the mean and standard deviation of n = 5–6 samples analyzed by two-way ANOVA with Tukey’s multiple comparisons test of one of three independent studies. Differences are significant (*p* < 0.05) when letters above bars are dissimilar. BMdEos—bone marrow-derived eosinophils; d—direct contact; id—indirect contact.

**Figure 7 cells-10-00509-f007:**
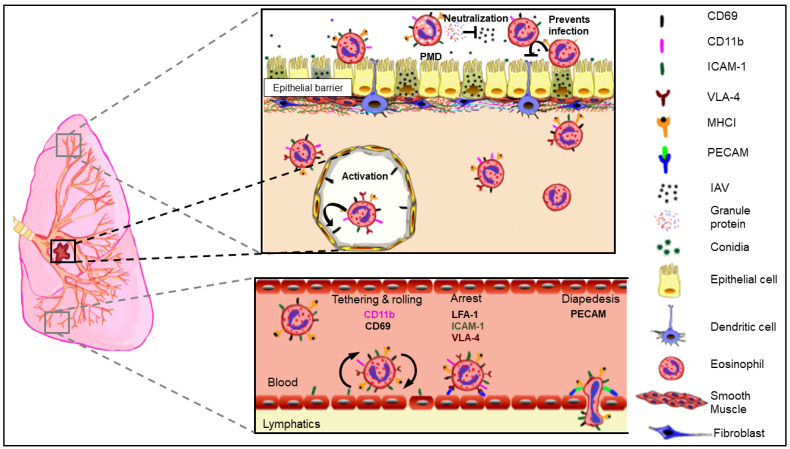
Schematic representation of conceptual model of eosinophil responses during influenza A virus (IAV) infection. Eosinophils may neutralize IAV directly or indirectly at the apical surface of the epithelial barrier. Infected eosinophils may engage in crosstalk with pneumocytes, thereby enhancing antiviral host protective mechanisms in these cells. Eosinophils activated by IAV may migrate to the lymphoid organs by upregulating surface expression of adhesion markers to interact with T cells.

## Data Availability

Microarray data have been deposited in the National Center for Biotechnology Information (USA) Gene Expression Omnibus database under accession number GSE163224.

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
