# Peer review of "Eosinophil Responses at the Airway Epithelial Barrier during the Early Phase of Influenza a Virus Infection in C57BL/6 Mice"

_cells, 2021, doi:10.3390/cells10030509_

Round 1

Reviewer 1 Report

The paper presents interesting findings that eosinophils host defense mechanisms in the early phase of influenza infection in a mouse model of allergic inflammation. The paper tests a novel hyporthesis in a numer of well-planned experiments, but some issues listed below need clarification and comment.

Major issues:

- cel line A549 grown in monolayer is not sufficient to confirm reduced virus-induced cytophatology – these findings should be recapitulated on primary epithelial cells in ALI culture model (pseudostratified epithelium) that better reflects in vivo situation. It should be at least mentioned as a study limitation in the Discussion section.

- the presentation of results section is not clear as the Authors frequently incorporate their previous results in this section. This make this section difficult to read and distinguish between already published data and the results from this study. The explanations of the results in the context of previous findings should be rather move to the Discussion section, whereas the Results sections should focus on summarizing and presenting the data obtained in this study. These two sections (Results and Disucssion) require comprehensive re-writing

- The Discussion section contains repetitions from the Introduction – these should be removed. Moreover, some paragraphs seem irrelevant in this section (e.g. description of influenza, lines 663-468; last paragraph, lines 575-589) as they are too general and too far from the main topic and the findings of this manuscript. I would suggest re-writing this section so it contains only information relevant for the findings of the manuscript, without over-interpretation of results

- the manuscript lacks conclusions (as the last section od the Discussion or as a separate section) that summarizes the main findings and their significance

- to confirm that eosinophils did not affect the epithelial barrier (the height of epithelial columnar cells) in naive mice that received eosinophils from the lungs of allergic mice, I suggest to add these data to the manuscript to validate these conclusions. Moreover, height of columnar cells is not the only and the best marker of epithelial barrier integrity and function. It is suggested to confirm this finding by measuring other markers e.g. barrier integrity marker E-cadherin and goblet cell metaplasia

Minor issues:

- in the Abstract the Authors mentioned ex vivo models, but they are not further mentioned in the manuscript. Could the Authors explain this discrepancy and correct the manuscript accoridingly.

- the manuscript contains numerous abbreviations throughout the manuscript and it makes it diffucult to read and understand the content as not all of them are explained. Similarly, full names of abbreviations should be added to the figure legends to make them self-explanatory

Author Response

Please see attached letter.

Reviewer 2 Report

Tiwary and colleagues investigated the role of eosinophils in infectious airway-models using a combination of exacerbating an asthmatic/allergic reaction followed by an influenza infection. They hypothesize that priming of the airways by an allergic reaction is beneficial for viral-clearance, potentially helping the barrier integrity.

The paper contains a lot of work and datasets, the experiments are well conducted in terms of statistics and analyses - however at points is a bit one-sided with the hypothesis and overly confident with some conclusions - in particular - given the non-tissue specific cell culture experiments, although the technologies are available.

I think the overall idea and findings are interesting, but would appreciate some revisions to the text and some experiments or clarifying comments.

The overall manuscript is well written and I can follow the authors logic and intentions. The experiments were conducted with significant power and according to standard, ethical guidelines.  The Figures are polished and look professional.

Comments

Introduction: Although the papers cited in the manuscript implicate a protective effect of asthma on infectious clearance - this is a divided topic. The provided science is underlining this train of thought - although for me personally it might be a chicken-egg question Particularly- given the flood of novel information on COVID19, I'd appreciate if this sentence could be removed from the manuscript. Current CDC guidelines in clinics and for the public still see asthma as a risk factor for infection, and without definitive proof in the manuscript - I do not see how this sentence elevates the value of the research at hand. 

Figure 1: I do not understand how the authors were able to exclude alveolar macrophages - which are high in SiglecF, and particularly after allergen induction can express CCR3 from their analysis. This either needs to be shown experimentally or clarified in the text.

Figure 2: This is a nice, self-explanatory Figure.

Figure 3: It is a bit unclear from the Figure/Text whether these lung eosinophils are initially recruited from the BM (like shown in previous figures with the BMEos-GFP) or whether these are resident eosinophils of the lung. A tail-vein injection experiment would be helpful to allow the reader to discern potential tissue-specific vs. circulatory effects. For the "cell-height" (3D) it would have been interesting to see how this compares to non-asthmatic animals, as the "height" could also indicate a shift of cellular composition or basal cell proliferation rate. 

Figure 4: This is where my main criticism starts for the paper. While the other aspects are small issues - the major issue I am having is the use of A549 cells and additionally the use of the Bone marrow eosinophils. If the coculture had happened with lung-derived eosinophils, that would have been potentially primed by the environment, or the use of primary lung epithelial cells (i.e. HBECs: readily available, and easily culturable with proprietary media), I could have accepted these results. However - given the lack of any tissue-specificity - I would like to ask the authors to provide either one - or both of these experiments/cocultures. Furthermore - dead cells are not equivalent to barrier integrity - barrier integrity and quality should be measured using TEER (transepithelial electrical resistance).

Figure 5: This would profit from some key-genes highlighted on the heat-map and some larger font on the String-networks.

Figure 6: Same comment as Figure 4 - the combination of BMEos and A549s is resulting in too arbitrary a system.

Figure 7: This is a beautiful Figure, that nicely highlights the ideas of the paper, but unfortunately is not backed-up by the experiments. I just wonder if some of the statements will withstand the challenge of time - given the arbitrary cell lines that were used. Also - given that the authors try to make claims about tissue-resident and circulatory cells here in this Figure- it becomes all the more important to add data, such as tail-vein injection for the FACS analyses and tissue-resident isolation of eosinophils to the cell-culture experiments to truly make these claims. 

Discussion: I agree with the authors conclusion about the eosinophil dynamics upon injury - but am not convinced by their statement about the improved barrier function.

I would also appreciate a bit more of a discussion on the discerning opinions, that asthma might not hold a protective function.

And the last point is again the covid point. I know that this is currently "a la mode", but given the massive difference between flu infections and corona infections, and the lack of Covid data in the paper, I'd appreciate any such grand claims to be removed.

I feel the overall manuscript has potential and has interesting findings,  I would be happy to review the revised manuscript.

Author Response

Please see attached letter.

Round 2

Reviewer 2 Report

The authors have significantly rewritten and reformatted the paper. They have highlighted the drawbacks of A549 cells.

Although no additional experiments into the direction of appropriate co-cultures were attempted, the toned-down text and reformatting of the focal point does reduce my criticism of the paper.

I still disagree with the Covid points - but it is a personal choice and the review process is meant to have a discussion, not force authors to delete anything. I do not insist on having it removed - it certainly provokes and thereby I guess the authors have achieved what they set out to.

Overall - clean manuscript, future experiments will show if things will hold true in more appropriate model systems. 

I do accept in present form.